# Bone Molecular Modifications Induced by Diagenesis Followed-Up for 12 Months

**DOI:** 10.3390/biology11101542

**Published:** 2022-10-21

**Authors:** Guillaume Falgayrac, Raffaele Vitale, Yann Delannoy, Hélène Behal, Guillaume Penel, Cécile Olejnik, Ludovic Duponchel, Thomas Colard

**Affiliations:** 1Univ. Lille, ULR 4490-MABLab- Adiposité Médullaire et Os, F-59000 Lille, France; 2CHU Lille, F-59000 Lille, France; 3Univ. Littoral Côte d’Opale, ULR 4490, F-Code postal Ville, 62200 Boulogne-sur-mer, France; 4Univ. Lille, UMR 8516-LASIRe-Laboratoire Avancé de Spectroscopie pour les, Intéractions la Réactivité et l’Environnement, F-59000 Lille, France; 5CNRS, UMR 8516, F-59000 Lille, France; 6Univ. Lille ULR 7367-UTML&A-Unité de Taphonomie Médico-Légale & d’Anatomie, F-59000 Lille, France; 7Univ. Lille, ULR 2694-METRICS: Évaluation des Technologies de Santé et des Pratiques Médicales, F-59000 Lille, France; 8University Bordeaux, CNRS, MCC, PACEA, UMR 5199, F-33600 Pessac, France; 9CHU Lille, Department of Oral and Maxillofacial Radiology, F-59000 Lille, France

**Keywords:** taphonomy, bone diagenesis, raman spectroscopy, collagen, mineral, cross-links

## Abstract

**Simple Summary:**

Improving knowledge about the mechanism of bone diagenesis is needed to help forensic investigators. Bone diagenesis refers to all processes (chemical or physical) that modify the chemical composition or the structure of skeletal remains. The mechanism of bone diagenesis is complex and poorly understood, especially at short timescales (between days and thousands of years). Identifying the chemical mechanism of diagenesis could help forensic investigators identify characteristics of skeletal remains, such as the sex, age or time elapsed since death. The aim of this work is to study bone diagenesis over 12 months on buried bone by Raman microspectroscopy. This technique allows for the evaluation of the chemical composition of bone. Human ribs from six individuals were buried for 12 months. The chemical composition of bone was analysed monthly by Raman microspectroscopy. The results showed that the mineral undergoes the dissolution–recrystallization mechanism. The collagen matrix of bone undergoes the hydrolysis mechanism. Hydrolysis induces the fragmentation of collagen by breaking the bonds that participate in the stability of collagen (cross-links). The results will help forensic investigators better understand the mechanism of bone diagenesis and the identification of skeletal remains.

**Abstract:**

After death, diagenesis takes place. Numerous processes occur concomitantly, which makes it difficult to identify the diagenetic processes. The diagenetic processes refer to all processes (chemical or physical) that modify the skeletal remains. These processes are highly variable depending on the environmental factors (weather, temperature, age, sex, etc.), especially in the early stages. Numerous studies have evaluated bone diagenetic processes over long timescales (~millions of years), but fewer have been done over short timescales (between days and thousands of years). The objective of the study is to assess the early stages of diagenetic processes by Raman microspectroscopy over 12 months. The mineral and organic matrix modifications are monitored through physicochemical parameters. Ribs from six humans were buried in soil. The modifications of bone composition were followed by Raman spectroscopy each month. The decrease in the mineral/organic ratio and carbonate type-B content and the increase in crystallinity reveal that minerals undergo dissolution–recrystallization. The decrease in collagen cross-linking indicates that collagen hydrolysis induces the fragmentation of collagen fibres over 12 months.

## 1. Introduction

After death, human remains undergo several processes of degradation, which vary strongly according to the location of the corpse [1]. Diagenetic processes are complex, especially in the early stages, because numerous processes take place. These processes are influenced by multiple factors that can be categorized into two groups: extrinsic factors (temperature, weather, soil, humidity, type of burial, scavengers, etc.) and intrinsic factors (age, sex, body weight, etc.) [2,3]. Bone diagenesis hampers the work of forensic investigators in the evaluation of skeletal remains, such as age, sex and postmortem intervals. Numerous studies have focused on the evaluation of diagenetic processes over long timescales (~millions of years) [4,5,6] but fewer over short timescales (between days and thousands of years) [7,8]. Bone diagenesis was evaluated by histology [9], chemistry [10] or physicochemical [11] approaches, allowing a better understanding of the complexity of the mechanism. Among the physicochemical approaches, vibrational spectroscopy (especially Raman spectroscopy) has shown its ability to provide decisive information on bone composition at the molecular level in physiological and pathological situations [12,13]. The assessment of the physicochemical parameters allows quantification of the modifications of the composition of bone. In the last decade, Raman spectroscopy has been increasingly employed to study skeletal remains under different conditions. Raman analysis is a nondestructive method, which allow for the perform of an additional technique on the same sample. Raman analysis provides access to the molecular composition of the mineral and organic matrix of bone simultaneously. The latter is a real advantage for the analysis of bone, since mineral and organic matrices are associated with complex 3D structures [14]. Moreover, the study of Trueman et al. on the diagenetic alteration of bone minerals by vibrational spectroscopy showed a correlation between the organic part and the crystallinity of the mineral in the early stage of the process [15].

Despite the ability of Raman spectroscopy to analyse mineral and organic matrices simultaneously, few studies have focused on the organic matrix of bone. Woess et al. (2017) evaluated the modification of composition by vibrational spectroscopy (reflection-, ATR- and Raman microspectroscopy). By analysing samples covering a forensic and archaeological period, they observed a reduction in the levels of phospholipids, proteins, nucleic acid sugars and complex carbohydrates [16]. Creagh et al. (2017) evaluated the modifications of composition by Raman and Fourier transform infrared microspectroscopy (FTIRM) on kangaroo and pig femurs over 120 days [17]. They also observed a loss of lipids over 120 days. Amadasi et al. (2017) found a low intensity of collagen bands in archaeological samples compared to forensic samples. McLaughlin et al. (2011) buried turkey leg bone over 68 days to study the modifications of bone by Raman spectroscopy. An increase in the intensity of the organic bands was monitored over burial time, which represented a counterintuitive result [18]. Additional experiments were performed by collagenase digestion to simulate microbial digestion. The additional results showed that the increase in organic bands was due, in part, to microbial digestion. Moreover, the effect of the extrinsic and intrinsic parameters was suggested in the cited studies, though few considered them in their evaluation. Baptista et al. (2022) showed that the effect of diagenesis on bone composition was different depending on age and sex [19]. In a recent study, our group quantified the effect of intrinsic and extrinsic parameters on Raman spectra, which demonstrated the need to take into account these parameters [11].

The objective of the study is to evaluate the molecular modifications induced by bone diagenesis by Raman microspectroscopy during the first 12 months.

## 2. Material and Methods

### 2.1. Samples

Six human subjects without known bone pathology were included in this work (Table 1). All subjects were Caucasian and died of heart attacks. To comply with ethical standards, the analysed bones were obtained from individuals who had “donated their bodies to science” according to a specific French law, which allows for anatomic dissections and research to be performed on these human cadavers. The patients gave informed written consent to participate, and the study was approved by an institutional review board: DC-2008-642. The ribs were chosen because they are also important in anthropology (e.g., estimating the age at death) [20,21]. It was decided to work with fresh bone samples as embalming procedures induce changes in their molecular composition [22]. Moreover, the use of fresh ribs permits the ability to mimic the conditions of the real-case scenario of the discovery of skeletal remains.

A repeated sampling protocol was chosen for this work. For each subject, two ribs (R1 and R4) were harvested the day of death. The flesh surrounding the bone was mechanically removed without any further treatment. A 5 mm thick section of bone was cut transversally on the proximal side of each rib using a diamond saw under water irrigation. The bone section was then used for Raman microscopy. This sample represented the baseline (M0). The outer proximal surface resulting from this preparatory step was then covered with neutral wax, before inserting the remaining ribs into their burial environment (Figure 1a–c). The wax is added to avoid the insertion of insect or soil when the rib is buried. The latter procedure was repeated each month for 12 months. One plastic bin was used per subject (Figure 2). The ribs (R1 and R4) of the same subject were placed in the centre of their own plastic bin. Each plastic bin (diameter = 60 cm, height = 30 cm) was filled with a clay soil typical of northern France (height of the soil = 15 cm). The pH was 6.8. The soil is described as brown to brown leached soils with little hydromorphism, aeolian silt on clay and sandy substrate of the Lille region. The ribs were covered by approximately 1 cm of soil. Plastic bins were placed outside and sheltered to protect them from scavengers and rain. The 6 plastic bins were located next to each other. This sampling method has been shown to not induce changes in the process of decomposition [23]. The local meteorological data were retrieved from the Metéo France website and are presented in Figure 3.

### 2.2. Raman Microspectrometry

Each 5 mm long section of ribs R1 and R4 was analysed by Raman microspectroscopy every month for one year. Raman spectra were acquired with a Raman microspectrometer LabRAM HR800 (Jobin-Yvon, Villeneuve d’Ascq, France) equipped with DuoScan technology, an XYZ motorized stage and a 785 nm laser diode [24]. With this technology, a mean spectrum is acquired over a rastered bone area of 30 × 30 µm^2^. The acquisition time was set at 30 s, and the spectral domain was 300–1700 cm^−1^. For each 5 mm thick section, the proximal transversal surface was stuck on a microscope slide. The distal transversal surface was polished with decreasing grain size (from 30 to 0.3 µm). For each rib, 4 zones were anatomically identified on the distal transversal surface: secondary osteon, interstitial bone, periost and trabecular bone, as shown in Figure 2e,f. Ten spectra were acquired per anatomical zone, which constituted forty spectra per rib and per month. The final set of 40 profiles was averaged prior to further processing.

The physicochemical variables (PPVs) were calculated from each Raman spectrum as follows: mineral/organic ratio was calculated from the area ratio of the v_1_PO_4_ (930–980 cm^− 1^) to CH_2_ (1434–1490 cm^− 1^) bands; crystallinity was calculated as the inverse of the full width at half maximum (FWHM) of the v_1_PO_4_ band; type-B carbonatation was calculated as the ratio between the respective area of the v_1_CO_3_ (1055–1090 cm^− 1^) and the v_1_PO_4_ bands; the hydroxyproline/proline ratio was calculated as the ratio between the intensity of their corresponding vibration bands centred at 871 cm and 854 cm^−1^; and the collagen cross-links were calculated using the intensity ratio between 1670 and 1690 cm^−1^, according to the method described by Gamsjaeger et al. [25]. The PPVs were calculated using MATLAB R2019a (MathWorks Inc., Natick, MA, USA).

### 2.3. Statistical Analysis

The 12-month evolution (expressed per month) of PPVs was estimated using linear mixed models including time as a fixed effect, with an unstructured covariance matrix to take into account the correlation between repeated measures within ribs and a random body effect to account for the correlation between ribs of the same body. Normality of the model residuals was checked using Q–Q plots. Statistical testing was performed at the two-tailed α level of 0.05. Data were analysed using the SAS software package, release 9.4 (SAS Institute, Cary, NC, USA).

## 3. Results

### Significant Variations of the Molecular Composition Were Observed in the Mineral and Organic Matrices as Function of Burial Time

Figure 4 shows averaged Raman spectra representative of the bone composition at baseline (M0) and after 12 months (M12). Each Raman spectrum is averaged over six bodies and both ribs. The Raman spectra at M0 and M12 show similar spectral profiles. The bands observed at 430, 585 and 960 cm^−1^ correspond to the vibrations of v_2_PO_4_, v_4_PO_4_ and v_1_PO_4_, respectively. The band at 1071 cm^−1^ is attributed to carbonate type-B (v_1_CO_3_). The other bands are assigned to proline (855 cm^−1^), hydroxyproline (879 cm^−1^), amide III (1245 and 1273 cm^−1^), CH_2_ (1450 cm^−1^) and amide I (1670 cm^−1^). Taken together, these Raman bands are characteristic of bone [26].

Modifications in bone composition were assessed through the evaluation of the five PPVs. Figure 5 shows the trends of the PPVs as a function of the burial time. Table 2 presents the results of the model. The slope of the mixed linear model is interpreted as the averaged variation of PPV per month over 12 months (ΔPPV). Four PPVs are significantly modified over the burial period (12 months). The mineral/organic ratio and the carbonation type-B significantly decreased at average rates of ΔPPV = −0.02 (*p* = 0.015) and ΔPPV = −0.03 × 10^−3^ (*p* < 0.001) per month, respectively. The crystallinity is significantly increased at the average rate of ΔPPV = +0.04 × 10^−3^ (*p* < 0.001) per month. The hydroxyproline/proline ratio is not modified over the burial period. The collagen cross-links were significantly decreased at an average rate of ΔPPV = −0.012 (*p* < 0.001) per month over 12 months.

## 4. Discussion

The objective of the study was to identify the molecular mechanism of bone diagenesis over 12 months through the evaluation of physicochemical variables (PPVs). The mixed linear model was chosen due to its ability to analyse the PPVs as a function of burial time, while considering the variability of both intrinsic parameters (rib and body source) in the analysis.

Over 12 months, the carbonation type-B decreases, and the crystallinity increases significantly, independent of the effect of the intrinsic parameters. These modifications, observed in previous studies, are typically described for archaeological remains [27,28]. Based on the literature, two hypotheses could explain the trends of carbonation type-B and crystallinity observed during the early stage of diagenesis. The first hypothesis is related to the presence of water in bone. Even when the ribs are buried in a dry (or low wet) environment in this study, bone is composed of water (up to 20% for trabecular bone and 10% for cortical bone’s wet weight) [29]. The water is present in microporosity of bone and also at the surface of the nanocrystals of hydroxyapatite (hydrated layer) [30,31]. The water will initiate the dissolution of hydroxyapatite. The carbonate is released during the dissolution, and it is known as the first molecule to be released during dissolution in carbonated hydroxyapatite [32]. After dissolution, the bone mineral matrix recrystallizes into a more thermodynamically stable crystal characterized by a higher atomic order (crystallinity increased) [1,33]. The second hypothesis is related to a process mainly driven by crystallization, which was proposed by de Sousa et al. (2020). They suggest that mineral dissolution has a minor contribution. Crystallization occurs by nucleation and precipitates in the free space left by the previous mineral or in the vascular structure [4].

The mineral/organic ratio decreased significantly by −0.020 (*p* = 0.015) per month over 12 months. A decrease in the mineral/organic ratio of bone was also reported in two studies involving animal models and burial periods of less than 24 months [17,18]. The evolution of diagenesis is known to be highly nonlinear [34,35] and influenced by a relatively numerous number of intrinsic and extrinsic factors [36], especially at short time scales. The decrease in the mineral/organic ratio implies a decrease in the mineral bands or an increase in the organic band. The decrease in the mineral band might be the consequence of the dissolution–recrystallization mechanism, as discussed in the previous paragraph. The increase in the organic bands is a less straightforward mechanism. A hypothesis could be based on an increase in the amount of collagen. In a previous work, our group showed by histology that the structure of the collagen matrix on the same samples was altered as a function of the burial time [9]. The alteration mechanism was identified as collagen hydrolysis. In a mathematical simulation, Collins et al. (1995) showed that collagen hydrolysis induces the fragmentation of collagen fibres [34]. The collagen fragments are not immediately cleared from bone due to their molecular weight [37]. Therefore, the increase in the intensity of the organic band (thus, the decrease in the mineral/organic ratio) is the consequence of the accumulation of collagen fragments in bone over the 12 months. An increase in the collagen bands was also observed in the work of McLaughlin et al. (2011) [18]. In comparison with previous studies concerning diagenesis at long timescales, the mineral/organic ratio is increased for archaeological samples when compared to modern bone [16,38,39]. The increase in the mineral/organic ratio might be related to the decrease in the intensity of collagen bands. Depending on the burial environment over millions of years, collagen is slowly cleared out from the bone and induces a decrease in the relative intensity of collagen, which is characteristic in archaeological samples [1,38,40,41].

The collagen matrix is analysed through the PPVs of the hydroxyproline/proline ratio and collagen cross-links. The collagen matrix is commonly described as the unit of an -X-Y-Gly-triplet, where -Gly- represents the glycine residues, and X and Y are often occupied by proline and hydroxyproline [42]. Hydroxyproline plays a crucial role in collagen stability during the formation of the triple helix. The hydroxyl group of hydroxyproline is essential for hydrogen bonding with water molecules. This interaction is important because the triple helix is maintained by a sheath of water molecules attracted by hydroxyproline [43]. Thus, hydroxyproline and proline are involved in the stabilization of collagen, by forming hydrogen bonds between collagen chains (intrafibrillar cross-links) [44,45]. In the current study, the hydroxyproline/proline ratio was not modified after 12 months. This result means that collagen hydrolysis does not alter the amount of hydroxyproline and proline. Thus, the intrafibrillar cross-links are not altered. The collagen cross-links were significantly decreased by −0.012 (*p* < 0.001) every month. The collagen cross-links provide an assessment of the collagen interfibrillar cross-links. Even if bones are buried in dry soil, bone contains water that is located in the microporous cavities of bone. The presence of water initiates the hydrolysis of collagen [30]. This result shows that the initial steps of the degradation of collagen is induced by hydrolysis, which cut the interfibrillar cross-links collagen. At a longer timescale, the hydrolysis cut the fibrils into shorter peptide units [30,34]. This finding is in agreement with the outcomes of the mathematical simulation of collagen hydrolysis performed by Collins et al. (1995) and their conclusions on the importance of collagen cross-linking in such a process [34].

This study has limitations related to the investigation of human samples. The experiments were performed with human ribs from six bodies. Their average age was old (mean age 83.5 years old). These choices were made as a compromise between the access to human bone material and the delay of the project. The trends observed in this study are mainly influenced by diagenesis, but the characteristics of human bone might also contribute.

In the literature, the PPVs related to the mineral matrix (PPVs-mineral) were evaluated as a function of the individual’s age. Yerrammshetty et al. (2006) evaluated the mineral/organic ratio, type-B carbonatation and crystallinity on the diaphysis of the human femora over six decades (52 to 85 years old) [46]. The results showed an increase in the parameters as a function of age and a reduction in the heterogeneity of the mineral bone composition. Similar results were observed by Gourion-Arsiquaud et al. (2009) on cortical bone from baboons (0–32 years old) [47]. An increase in the mineral/organic ratio and type-B carbonation was observed as a function of the baboon age. Thus, young individuals have a lower PPVs-mineral matrix, and the mineral bone composition is more heterogeneous than that of old individuals. In our study, ribs from the old population were used. Variations in the PPVs-mineral induced by diagenesis observed in our study would be different with ribs from younger individuals (e.g., lower amplitude of variation or higher variability).

The origin of the bone might also influence the results. The composition of bone differs depending on the location and function [48,49]. For example, the amount of cross-links is a function of the location and function of bone due to its relation with bone turnover [43]. Thus, the trends observed in our study would be different on a bone with another function, such as a mandible or a femur.

The design of the protocol in this study (repeated sampling) implies that each rib is exhumed and buried 12 times. This repeated handling could have induced perturbations in the diagenetic changes. As an alternative protocol, an elimination protocol was considered to limit repeated handling. The elimination protocol implies the use of 12 ribs buried the same month. Then, each month, a new rib is exhumed, analysed and eliminated for 12 months. Even if the elimination protocol limits the handling of the same rib, it would induce variability associated with the origin of the ribs (individual and/or location). This additional source of variability could hinder the variability induced by the diagenetic changes [50]. Moreover, the study of Adlam et al. (2007) compared both protocols on bone material [23]. The decomposition of bone was quite similar between both protocols. Considering these implications and the aim of our work, the repeated sampling protocol was preferred over the elimination protocol.

## 5. Conclusions

The objective of this work was to evaluate the molecular modifications at the early stage of the diagenetic processes by Raman microspectroscopy. The early stage of diagenesis was monitored over 12 months of burial time on human bone samples. The use of fresh ribs permits the mimicry of the conditions of a real-case scenario of the discovery of skeletal remains. The effect of intrinsic factors was taken into account by the use of a linear mixed model. A significant modification of the molecular composition was observed over the burial time, in agreement with previous studies. Over 12 months, the mineral goes through a mechanism of dissolution–recrystallization, which is characterized by a decrease in the mineral/organic ratio and carbonate type-B content and an increase in the crystallinity. Collagen hydrolysis induces the fragmentation of collagen fibres, which is characterized by a decrease in collagen cross-links. In our burial conditions, the diagenetic mechanisms of human bone are governed by dissolution–recrystallization mechanisms for the mineral matrix and collagen hydrolysis for the organic matrix. This study provides new details on early-stage diagenetic mechanisms to help the forensic community.

## Figures and Tables

**Figure 1 biology-11-01542-f001:**
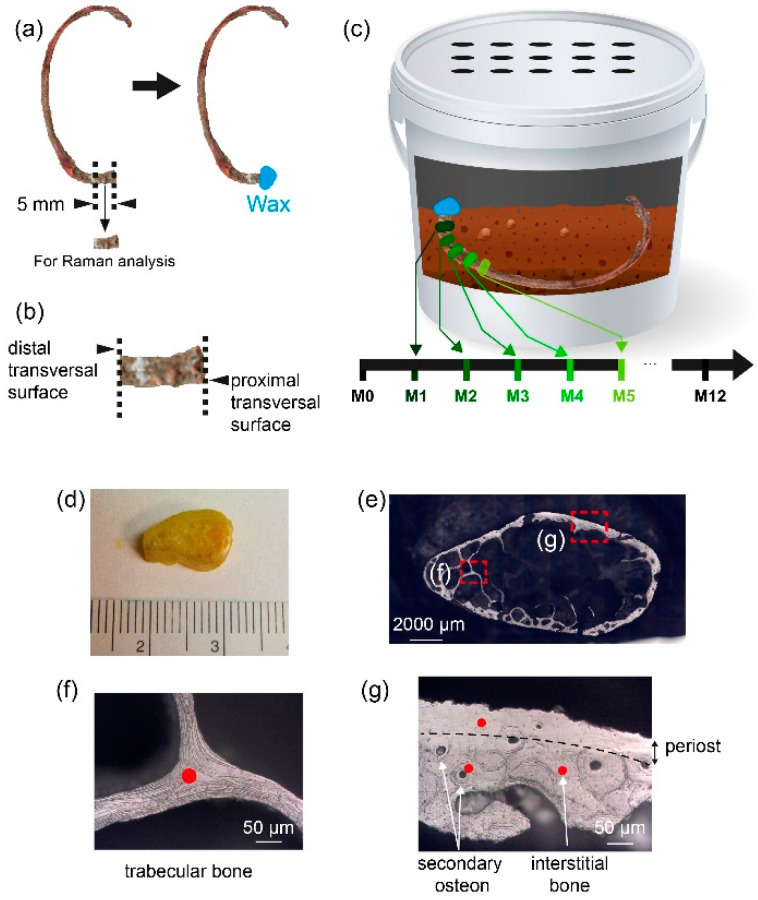
Description of the protocol of preparation of the rib for Raman analysis. This protocol is repeated 12 times for 12 months for each rib. (**a**) Left panel: schema of a rib before the cutting procedure at M0. The dashed lines represent the limits of the cut. Right panel: rib after cutting procedure. The wax protects the outer proximal surface. (**b**) Zoom on the sample analysed by Raman microspectroscopy. The proximal transversal surface is protected by the wax. The distal transversal surface is not in contact with the wax and is analysed by Raman microspectroscopy. (**c**) Representation of a plastic bin with soil and one rib. One rib is represented to simplify the representation. It shows that a new 5 mm thick section of the same rib is cut each month and analysed by Raman. To do the cutting procedure, the rib is exhumed and reburied each month. (**d**) Piece of rib after the cutting procedure. The visible side corresponds to the distal transversal surface. (**e**) Optical image of the piece of rib under Raman microscope. The visible side corresponds to the distal transversal surface analysed by Raman microspectroscopy. The insets (**f**,**g**) show examples of each type of bone and the area analysed by Raman microspectroscopy. (**f**) Example of trabecular bone analysed. (**g**) Example of cortical bone showing periost, secondary osteon and interstitial bone analysed. The red dot represents the localization of the analysis. For each type of bone, 10 spectra were acquired at 10 different locations.

**Figure 2 biology-11-01542-f002:**
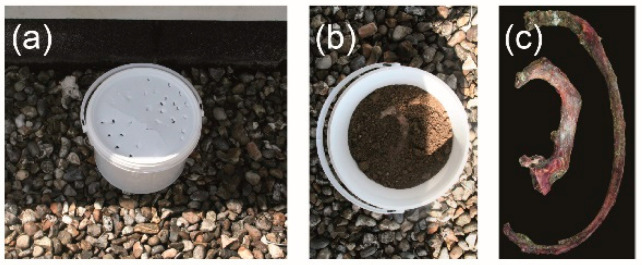
(**a**) Plastic bin with lid to protect from scavengers and rain; (**b**) plastic bin filled with clay soil typical of northern France carrying the rib. The rib is covered by 1 cm of soil; (**c**) ribs nos. 1 and 4 after 1 month of burial time.

**Figure 3 biology-11-01542-f003:**
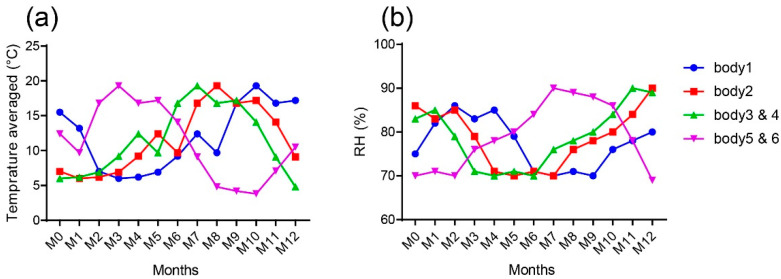
(**a**) Averaged outdoor temperature (°C) per month during 12 months for each body. (**b**) Averaged RH (%) during the 12 months for each body. The month M0 is different depending on the body. For body 1, M0 = September 2013; body 2, M0 = November 2013; bodies 3 and 4, M0 = December 2013; bodies 5 and 6, M0 = April 2014.

**Figure 4 biology-11-01542-f004:**
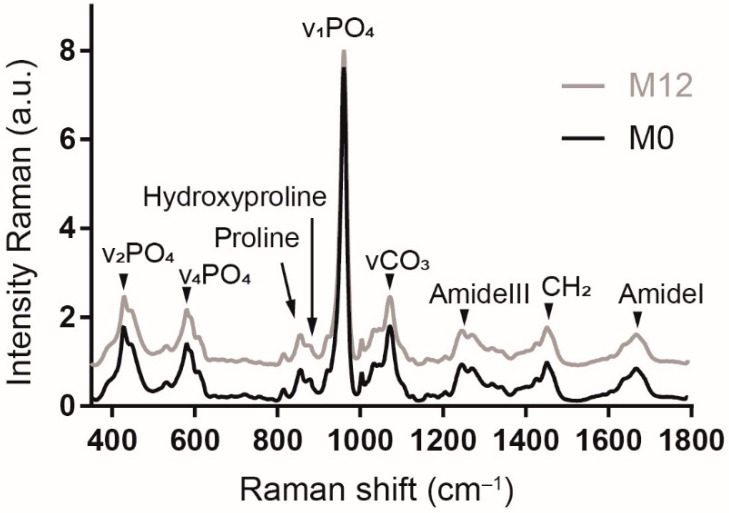
Comparison of averaged Raman spectra at baseline (M0) and after 12 months (M12). Each Raman spectrum is averaged over the 6 bodies and both ribs. The spectrum M12 is shifted along the *y*-axis to facilitate the comparison of M0 and M12. The shift does not have a biological or instrumental significance.

**Figure 5 biology-11-01542-f005:**
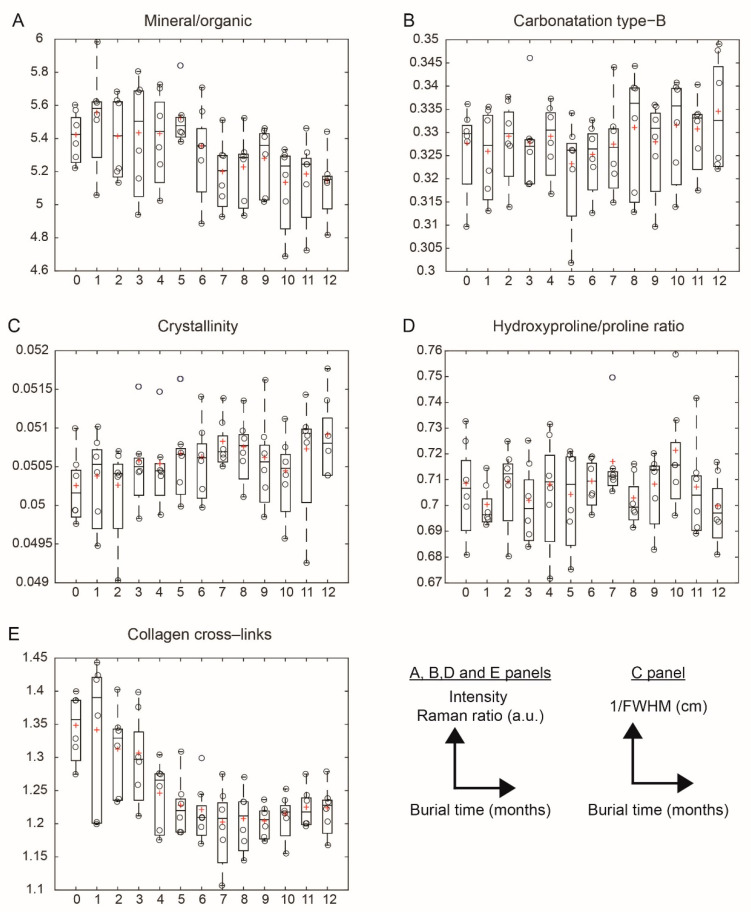
Evolution of the 5 physicochemical variables (PPVs) as a function of burial time in months. One boxplot represents the variations in one parameter at one month. Each circle represents the mean value of a parameter of one body. The red cross represents the mean value of a parameter over the 6 bodies. The horizontal line represents the median value over the 6 bodies. (**A**) Mineral/organic ratio; (**B**) carbonatation type-B; (**C**) crystallinity; (**D**) hydroxyproline/proline ratio; (**E**) collagen cross-links.

**Table 1 biology-11-01542-t001:** Characteristics of the subjects: 4 males and 2 females; mean age of the group 83.5 (±7.1) years old.

ID	Sex	Age	Observations
#1	male	72	no records
#2	female	92	coxarthrosis, knee surgery, osteonecrosis of the femoral head
#3	female	80	heart prosthetic stent, vascular issues, Alzheimer’s
#4	male	88	hypertension
#5	male	82	acute myeloid leukaemia, alcoholism, smoker
#6	male	87	no records

**Table 2 biology-11-01542-t002:** Physicochemical variables evaluated over the 12 months of burial time across the 6 subjects and the 2 ribs. The slope estimated by the linear mixed model is the averaged variation in PPV per month over 12 months (ΔPPV).

Variables	ΔPPV (95%CI)	*p* Value
Mineral/organic	−0.020 (−0.036 to −0.004)	0.015
Carbonation type-B	−0.35 × 10^−3^ (−0.48 × 10^−3^ to −0.23 × 10^−3^)	<0.001
Crystallinity	+0.04 × 10^−3^ (+0.02 × 10^−3^ to +0.06 × 10^−3^)	<0.001
Hydroxyproline/proline	+0.28 × 10^−3^ (−0.49 × 10^−3^ to 1.05 × 10^−3^)	0.47
Collagen cross-links	−0.012 (−0.014 to −0.009)	<0.001

## Data Availability

The data presented in this study are openly available at the Mendeley repository at http://dx.doi.org/10.17632/ymg8drmby7.1, accessed on 14 September 2022.

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
