# Peer review of "Bone Molecular Modifications Induced by Diagenesis Followed-Up for 12 Months"

_biology, 2022, doi:10.3390/biology11101542_

Round 1
Reviewer 1 Report (Previous Reviewer 4)
The Authors have made a great effort to improve this revised version of the manuscript. The Authors have addressed the comments and suggestions offered by the reviewers and the manuscript has significantly improved. In addition, more figures and tables were added, improving the interpretation of the experimentation and the results. The manuscript can be accepted for publication into Biology. In the following please find a relatively minor and specific suggested change:
*Line 21. Change “aspect” to “structure”.
Author Response
Please see the attachment

Reviewer 2 Report (New Reviewer)
The paper by Falgayrac and colleagues has been executed with care, and what follows are comments meant to help the authors improve compression and understanding.
1. There is a disjunct between the claim on line 64-65 that the Raman analyses require little or no preparation, and the sawing, grinding, and polishing protocol described in the methods. Considerable preparation to obtain a flat polished surface for Raman imaging is reasonably (an appropriately) destructive.
2. Mention that removing the flesh from around the bone (line 111) is atypical of a forensic situation. The literature may be somewhat unsettled on the matter, but removing tissue may affect the potential that endogenous bacteria have to contribute to diagenesis.
3. I am not certain that I understood the protocol completely. Please confirm: the 5 mm thick block was sawn free from the proximal (according the to picture) end of the rib, the wax was adhered to the proximal-most side of the block, the bone was buried and exhumed each month, ground and polished, and then imaged. If that is correct, then no need to modify the text, but if not, please make clearer.
4. Please explain in more detail the purpose of placing wax on one surface.
5. I found the “non-exposed” and “exposed” terms confusing. I rather thought that the non-exposed surface is that surface exposed to the soil, and subsequently modified and imaged. It seems to me that the term “exposed” should be given to that surface experiencing contact with the environment.
6. It is highly useful that the authors kept the mineral intact versus removing the mineral to analyze the collagen, as was done in reference [9] of the paper, permitting a more holistic perspective. But then there is no interpretation of the findings in relation to real-world phenomena common in the forensic literature concerning bacterial, fungal, and other environmental phenomena (e.g., soil pH). Soil pH was reported, but then not employed in any context. Two things to address: 1) Collagen degradation is well described in the paper, except for explaining in some detail the processes involved in hydrolysis of the collagen. 2) It will be helpful to have informed discussion regarding the removal of the carbonate and increased crystallinity of the study samples. Might the soil pH have been acidic enough, or might bacterial processes have contributed; there is no mention in the paper, but the soil matrix was not sterilized and bacteria were surely present. Were there no histological indications of bacteria or fungi?
Author Response
Please see the attachment

This manuscript is a resubmission of an earlier submission. The following is a list of the peer review reports and author responses from that submission.
Round 1
Reviewer 1 Report
The paper is correct and provides new information of early bone diagenesis in an experiment performed by the authors by a year. I find it correct and suitable for the Historical Biology, results agree with previous publications and this is something to take into account given the use of new methodologies.
I agree it can be published as it is although I found small spelling mistakes: line 225: Hydroxyproline playS a crucial role for collagen and line 254: The collagen hydrolysis induces the fragmentation of the collagens fibres. I don't feel qualified to correct the English editing, but a quick reading should be done.
Author Response
Please see attachement

Reviewer 2 Report
The topic is very important, and gains much interest from readers. Considering the importance of the topic discussed in the manuscript, the postmortem interval for forensic practice, the main focus of the review was on the meaning of this paper for forensic practice, which requires high standards. The analytical technique chosen, and chosen PPVs are interesting and substantiated. Unfortunatly, several major flaws were found leading to the conclusion that this publication has little to no value for forensic practice.
Major points:
- The authors state in the first paragraph, line 41-42, that there is no reliable method for determining the PMI, please elaborate and provide substantiation from literature, because for skeletal remains there is radiocarbon dating, which is very reliable.
- i.e. but not limited to line 49; forensic period, please elaborate, how long is such a forensic period? Same goes for the archaeological period. Please provide substantiation, and is there no overlap?
- Lines 94-95/120/158/170/173: Error! Reference source not found. This has to be fixed.
- Lines 96-97: please add a section on the ethical and legal framework in place in France on the donation program and institute that the bodies were donated to, and if applicable information on the ethical approval or procedure from the institute that provided the samples at the end of the manuscript.
- Sample population; old age category, ribs, these choices (perhaps inevitable) influence results, there could be more elaboration in the discussion on this, besides mentioning the small sample size.
- Line 108> The authors explain that a transverse bone section was taken, 5mm long, does that mean thickness? Please clarify (also line 126). Then on Line 111 the authors refer to a hole, this is unclear, is the hole a reference to the exposed internal structure of the rib due to the transverse section? If so, because ribs do not have a medullary cavity, hole might not be the right word, exposed transverse surface would be clearer (if that was the case).
- Methodology, lines 112-113, does this mean the same sample was exhumed twelve times and each time a transverse section was taken and the rib reburied. This should have been an elimination study instead of repeated sampling. Each time the rib is sampled the burial context is disturbed, this has major implications on the results and the translation to practice. This is not discussed in the current manuscript and has to be addressed.
- Methodology, the section was stuck on a microscopic slide, polished and then measured with raman, please elaborate on what surface was measured with raman; the outer periosteal surface of the rib or the exposed transverse section surface. This is very important to clarify, measuring repeatedly the freshly exposed surface hampers the analysis, after each month the exposure and thus changes begins from almost zero. This could explain the huge variance seen in fig 2 > see point below. If the above was not the case, then the authors should have made more clear how the sampling took place, i.e. by means of a figure.
- Methodology, how was dealt with contamination from cutting the section and fresh unexposed bone sticking to the surface that was measured with the raman? Were the samples degreased? Was there any debris left on the bone from polishing the slide? Normally there is, grit from the sandpaper (every contact leaves its trace).. Contamination, bone dust, has a huge influence on sensitive analytical techniques, like raman.
- Results: figure 1 shows an increase in arbitrary unit (a.u.) over the whole spectrum, it looks like the base was increased. It is unclear to me when M12 has a higher output over the whole line (all PPVs) than M0, how can figure 2 then show a decrease in ratios, when both values to calculate the ratio increased. Besides, was it not expected that some values (PPVs) to decrease over time, like the amides instead of gaining a higher a.u.? Lastly, was there a calibration carried out for the a.u.? This is maybe not necessary for the ratio’s, but for the raw data it seems to be relevant.
- With regard to the statistical analysis, deltaPP was calculated for the LinearMixedModel, but no graphs are presented of the deltaPP. Fig 2 is raw data in a boxplot. This raw data shows a huge variance and thus does not support the significant findings based on the deltaPP in table 3. The discussion does not cover the huge variance shown in fig 2 for any of the studied PPVs. The current statistical approach is very limited, it has no meaning for practice because the significances do not apply to practice at all (also see comment on methodology repeated measurements instead of a elimination experiment), variance should have been taken into account.
Minor points:
- Gender = sex (line 24), please check manuscript and make changes throughout.
- Diagenesis is used instead of diagenetic, i.e. in the abstract “evaluated bone diagenesis processes over..” should be “diagenetic processes over..” (line 25). Otherwise use Diagenesis without processes, i.e. (line 26) : “early stages of diagenesis processes by Raman” should read “early stages of diagenesis by Raman”. Please check manuscript and make changes throughout
- In more than one occasions in the manuscript there is a present tense, i.e. (line 26) “The objective of the study is to assess” should read “The objective of the study was to assess”. Please check manuscript and make changes throughout. (also lines 121, 178)
- The abstracts contains sentences that are exactly the same as in the manuscript, please rewrite.
- Line 37: it is unclear what the authors mean with “delay” between death and discovery, please rewrite.
- Line 52, provide substantiation and cite the methods that are now mentioned, so one reference for the method one for the publication that proofs its unreliability, if not the same.
- Line 73: “Creagh et al”, should be written as follows “Creagh et al. (year) evaluated..”. Please check manuscript and make changes throughout.
- Abbrevations should be written out the first time they are used in the manuscript, i.e. but not limited to FTIRM, line 74. Please check manuscript and make changes throughout.
- Table 2, Environmental conditions, under cover is not very clear. Subsurface is what it is referred to in other taphonomic studies. Soil type, is this in accordance with the “United States Department of Agriculture, textural classification triangle”?
- Line 111, T0, and line 159 M0, I believe the authors mean the same but this is now unclear.
Reviewer 3 Report
The article highlights the problems in evaluating the early stages of diagenesis processes by Raman and focuses on the analysis of samples covering a forensic time period. I appreciate that the study was conducted on fresh ribs, which allows it to mimic the conditions of a real case of skeletal remains discovery. The article points out the limitations, such as the small sample size. The results of the present study seem to be a useful tool for the purposes of forensics. Nevertheless, there are some points that should be revised and discussed in more detail.
The major recommendations:
- The aim of the present study is not entirely clear. Do the authors primarily want to point out the use of Raman microspectroscopy? If so, this should be somehow highlighted (advantages/disadvantages), compared and discussed, especially in the discussion section. Then the authors mention that the study could be useful in the forensic field, but this also needs to be discussed in more detail. The results confirm that the decrease in mineral/organic ratio means a decrease in mineral bands or an increase in organic bands. Are the results different from those covering the archaeological period? If the authors want the study to be understood as a forensic contribution, it is essential that this be compared or at least considered in the discussion section. For example, a valuable study demonstrated that a francolite crystal can distinguish between ancient and recent bone.
Throughout the manuscript, e.g., in the Introduction and in the Material/Method section, there are many places where there is an "Error! Reference source not found." I suggest revising this and using the appropriate source.
The minor suggestions:
- I assume that the temperature and pH were approximately the same throughout the 12-month period (as stated in the Material/Method section). This should be added. It is well known that temperature and pH are important factors affecting collagen stability.
- Since the ribs were removed and placed in their environment to obtain a 5-mm section, the possible diagenetic change should be considered in the study, even if the long-term diagenetic change should not be present.
Reviewer 4 Report
This study explores molecular changes at the early stage of the diagenesis process using Raman spectroscopic data to provide important information regarding the estimation of post-mortem interval. The study sample includes first and fourth ribs from six human subjects. The results show that diagenetic process is governed by dissolution-recrystallisation mechanisms for the mineral matrix and collagen hydrolysis for the organic matrix. This is a very interesting study on an important parameter in forensic research: the estimation of post-mortem interval in human skeletal remains.
Overall, the manuscript is well-structured and well referenced. However, this study shows specific weakness which should be addressed.
MAJOR REVISIONS
MATERIAL AND METHODS. What is the dimension (diameter and height) of the plastic bin? How much volume of soil was put in the plastic bin? Was the rib placed in the center of the plastic bin or near the walls? Was one plastic bin used per bone or was a single bin used for all bones? In the case of multiple bins, were they located next to each other or separated by a certain distance? Was the plastic bin kept open or closed? It is necessary to provide more details on these aspects to better understand the experimentation.
MATERIAL AND METHODS. Table 2 appears to show only the data (temperature and humidity) from a single bin containing ribs R1 and R4 in a single time period (probably t0). This table should show the temperature and humidity data of each of the ribs of each of the bins in each of the analysed time periods (t0, t1, t2, t3, etc.). It is necessary to know these parameters to be able to observe how the changes observed with the spectroscopic data may or may not be related or affected by changes in temperature and moisture over time. I don’t think that the experimentation has performed in an outdoor place with a “stable” climatic condition all year round, so I can imagine that the temperature and humidity have changed over time during the day/night and during the different months and seasons.
MINOR REVISIONS:
*Lines 24, 84 and Table 1. The variable “gender” is used as synonym of “sex”. Sex and gender are not synonyms. “Sex” is a biological term. It refers to the biological differences between males and females, such as the genitalia and genetic differences. “Gender” is more difficult to define, but it can refer to the role of a male or female in society (gender role) or an individual’s concept of themselves (gender identity). Please, change these terms as they are not interchangeable.
*Line 26. Change the term “microspectrocopy” to “microspectrocopy”.
*Line 27. “[…] Raman microscpectrocopy during the 12 months.” Remove “the”.
*Line 43. Remove “skeletal”.
*Lines 88–91. “The mineral and […] extrinsic parameters”. This paragraph corresponds to Material and Methods. Remove these sentences from here.
In lines 95, 120, 158, 170, 173. In some parts of the text the following error message appears: “Error! Reference source not found”. Please check all the text.
*Line 104. Change “characteristics” to “Characteristics”.
*Line 167. Change “comparison” to “Comparison”.
*Line 200. Change “molecules” to “molecule”.
